# Combined Effect of Vacuum Packaging, Fennel and Savory Essential Oil Treatment on the Quality of Chicken Thighs

**DOI:** 10.3390/microorganisms7050134

**Published:** 2019-05-15

**Authors:** Miroslava Kačániová, Martin Mellen, Nenad L. Vukovic, Maciej Kluz, Czeslaw Puchalski, Peter Haščík, Simona Kunová

**Affiliations:** 1Department of Microbiology, Faculty of Biotechnology and Food Sciences, Slovak University of Agriculture, 949 76 Nitra, Slovak Republic; 2Department of Bioenergy Technology and Food Analysis Faculty of Biology and Agriculture, University of Rzeszow, 35-601 Rzeszow, Poland; kluczyk82@op.pl (M.K.); cpuchal@ur.edu.pl (C.P.); 3Klas Holding, s.r.o., 984 01 Lučenec, Slovakia; martin.mellen@gmail.com; 4Department of Chemistry, Faculty of Science, University of Kragujevac, P.O. Box 60, 34000 Kragujevac, Serbia; nvchem@yahoo.com; 5Department of Technology and Quality of Animal Products, Faculty of Biotechnology and Food Sciences, Slovak University of Agriculture, 949 76 Nitra, Slovak Republic; peter.hascik@uniag.sk; 6Department of Food Hygiene and Safety, Faculty of Biotechnology and Food Sciences, Slovak University of Agriculture, 949 76 Nitra, Slovak Republic; simona.kunova@uniag.sk

**Keywords:** *Foeniculum vulgare*, *Satureja hortensis*, chicken thigh, microflora

## Abstract

The aim of the present work was to evaluate the microbiological quality of chicken thighs after treatment by fennel (*Foeniculum vulgare*) and savory (*Satureja hortensis*) essential oil, stored under vacuum packaging (VP) at 4 ± 0.5 °C for a period of 16 days. The following treatments of chicken thighs were used: Air-packaging control samples (APCS), vacuum-packaging control samples (VPC), vacuum-packaging (VP) control samples with rapeseed oil (VPRO), VP (vacuum-packaging) with fennel essential oil at concentrations 0.2% *v*/*w* (VP + F), and VP with savory essential oil at concentration 0.2% *v*/*w* (VP + S). The quality assessment of APCS, VPC, VPRO, VP + F and VP + S products was established by microbiological analysis. The microbiological parameters as the total viable counts of bacteria of the *Enterobacteriaceae* family, lactic acid bacteria (LAB), and *Pseudomonas* spp. were detected. Bacterial species were identified with the MALDI-TOF MS Biotyper. The combination of essential oils and vacuum packaging had a significant effect (*p* < 0.05) on the reduction of total viable counts (TVC) compared with control group without vacuum packaging and the untreated control group. Though 15 genera and 46 species were isolated with scores higher than 2.3 from the chicken samples.

## 1. Introduction 

The number of reported cases of food - associated infections has increased rapidly so it is very important to focus on food safety from both consumers and food industry point of view [1]. One of very popular food commodity in recent decades in many countries is poultry meat, which is a highly perishable food providing an almost perfect medium for microbial growth, including both spoilage and pathogenic microorganisms [2,3]. One of the possibilities to prolong the shelf-life of the meat is application of natural preservatives, including oil extracts of herbs and spices [4]. The oil s are natural products extracted from plants and can be used as natural additives for food because of their antibacterial, anti-fungal, antioxidant and anti-carcinogenic properties. [5,6,7]. Essential oils have been proven to show an inhibitory effect against a wide range of food spoiling microbes; however, the level of this effect depends on their concentration, method of testing, and the active constituents presented [8,9]. The antimicrobial activity of essential oils is the consequence of the presence of small terpenoids and phenolic compounds (thymol, carvacrol, eugenol) [10,11,12].

*Foeniculum vulgare* Mill. (fennel) is a biennial medicinal plant belonging to the family *Apiaceae*. The essential oil of fennel is used as flavoring agents in food products. The main components of fennel essential oil are *trans*-anethole, fenchone, and estragole [13,14,15]; among them, *trans*-anethole exhibits both antioxidants and antimicrobial activities [16]. In addition, Patra et al. [17] reported that anethole and its isomers are responsible for the antimicrobial activities of fennel oil.

*Satureja hortensis* L. (summer savory) is a well-known aromatic and medicinal plant. Some parts of the *Satureja hortensis* plant such as leaves, flowers, and stem are frequently used for tea or as an additive in commercial spice mixtures for a lot of food and give the food the aroma and flavour [18,19,20,21]. Thymol, p-cymene, γ-terpinene, and carvacrol are the main components of savory oil, exhibiting strong biological and antimicrobial activity [22,23,24,25]. 

In this study, we aimed to investigate the effect of (*Foeniculum vulgare*) and savory (*Satureja hortensis*) essential oils on the shelf-life extension of fresh chicken thighs stored under vacuum packaging at 4 ± 0.5 °C for a period of 16 days. 

## 2. Material and Methods

### 2.1. Preparation of Essential Oils for Testing and Their Chemical Composition

The dried medicinal plants for essential oil isolation were donated by established growers. Essential oils were distilled in the large-scale distillation apparatus specifically designed for aromatic and medicinal plants. The apparatus consisted of the main distillatory apparatus, a steam condenser, a steam boiler, and an apparatus for the improvement of used water. There were two types of equipment—an HV-3000 with a height of 5250 mm, a width of 2180 mm, and a container for 200–250 kg of dried plant material and an HV-300 with a height of 3400 mm, a width of 1300 mm, and a container for 40–50 kg of dried plant material. An analysis of the essential oils was carried out with Hewlett-Packard 5890/5970 GC-MSD system.

A chemical composition of the essential oils *of Foeniculum vulgare* Mill. was: t-anethol (32.8%), and of essential oil *Satureja hortensis* were carvacrol (45.2%), γ-terpinene (30.5%), α-terpinene (3.2%), and p-cimene (2.3%).

### 2.2. Preparation of Samples

Rapeseed oil with any microorganisms was used as a solvent for concentration of essential oils. Rapeseed oil has good properties, and it is safe for human health.

A vacuum packaging machine—type VB-6 (RM Gastro, Czech Republic)—was used for packaging. Fennel and savory essential oils (Calendula, Nova Lubovna, Slovakia) were transferred onto chicken breast with a micropipette to completely cover the surface of both sides of the meat. 0.2% *v*/*w* of fennel (*Foeniculum vulgare*) and savory (*Satureja hortensis*) essential oils were used for treatment. Both essential oils were prepared for a concentration of 0.2 % *v*/*w* in rapeseed oil (*Brassica napus*) purchased in a Slovak market.

Fresh chicken thigh meat (cca. 300 g, skinless and boneless fillet) was purchased from a local poultry processing factory within 1 h after slaughter and transferred to the laboratory of Slovak University of Agriculture, Nitra, Slovakia, in insulated polystyrene boxes on ice. The chicken meat was kept at 4 °C until the testing was initiated. In each group were three samples for microbiological analysis. Each experiment was replicated twice. The fresh chicken thigh samples were prepared as follows:

Air-packaged control samples (APCS): Fresh chicken thigh meat was packed into polyethylene bags (bags for food packaging with low permeability, as well as gas and vapor protection) and stored aerobically at 4 °C.

Vacuum-packaged control samples (VPC): Fresh chicken thigh meat was packed into polyethylene bags and stored at 4 °C anaerobically in the vacuum.

Vacuum-packed control samples (VP) with rapeseed oil (VPRO): Fresh chicken thigh meat was treated with rapeseed oil for 1 min, packed in the polyethylene bags, and stored at 4 °C anaerobically in the vacuum.

VP with fennel oil 0.20% *v*/*w* (VP + F): Fresh chicken thigh meat was treated with fennel oil for 1 min, packed to polyethylene bags, and stored at 4 °C anaerobically in the vacuum.

VP with savory oil 0.20% *v*/*w*, (VP + S): Fresh chicken thigh meat was treated with savory oil for 1 min, packed to polyethylene bags, and stored at 4 °C anaerobically in the vacuum;

Each sample was packaged immediately after treatment by using a vacuum packaging machine—type VB-6 (RM Gastro, Czech Republic)—and kept at 4 °C ± 0.5 °C for 16 days. 

### 2.3. Microbiological Analysis

10 g of chicken thigh was sampled with sterile scalpels and forceps and immediately transferred into a sterile stomacher bag containing 90 mL of 0.1% buffered peptone water (BPW, pH 7.0, Oxoid, Basingstoke, UK); the sample was then homogenized for 60 s in a stomacher at room temperature. Sampling was carried out at on day 0, 4, 8, 12, and 16 of the experiment

For each sample, appropriate serial decimal dilutions were prepared in 0.1% BPW solution. The amount of 0.1 mL of serial dilutions of prepared samples was spread on the surface of dry media. Total viable counts (TVC) were counted on a Plate Count Agar (PCA, Merck, Darmstadt, Germany) after incubation for 3 days at 30 °C, aerobically. The number of pseudomonads were determined on a Cephaloridine Fucidin Cetrimide agar (Oxoid, supplemented with SR 103, Basingstoke, UK) after incubation at 25 °C for 2 days, aerobically. For the detection of *Enterobacteriaceae*, 15 mL of molten (45 °C) Violet Red Bile Glucose Agar (Oxoid) was inoculated with 1.0 mL of the sample. Incubation was carried out at 37 °C for 24 h, aerobically. The number of lactic acid bacteria (LAB) were determined on a Man Rogosa Sharpe agar (Oxoid) after incubation at 25 °C for 5 days, anaerobically. Then, the agars were evaluated for bacterial growth, and 8 colonies (depending on the different morphological characteristics of colonies) per plate were selected for further confirmation with the MALDI-TOF MS Biotyper (Bruker Daltonics, Germany). 

### 2.4. Mass Spectrometry Identification of Isolates

The qualitative analysis of microbial isolates was performed with MALDI-TOF Mass Spectrometry (Bruker Daltonics, Germany). Isolates from the agar were transferred into 300 μL of distilled water. Then, a quantity of 900 μL of ethanol was added, and the tubes with bacterial suspension in water were centrifuged for 2 min at 14,000 npm. The supernatant was discarded, and the pellet was centrifuged repeatedly. After the remaining ethanol was removed, the pellet was allowed to dry. An amount of 10 μL of 70% formic acid was mixed with the pellet, and a 10 μL of acetonitrile was added. Tubes were centrifuged for 2 min at 14,000 npm, and 1 μL of the supernatant was used for MALDI identification. Once dry, every spot was overlaid with 1 μL of the α-Cyano-4-hydroxycinnamic acid (HCCA) matrix and left to dry at room temperature before analysis. Generated spectra were analyzed on a MALDI-TOF Microflex LT (Bruker Daltonics) instrument using Flex Control 3.4 software and Biotyper Realtime Classification 3.1 with BC-specific software. Criteria for reliable identification were a score of ≥2.0 at species level and ≥1.7 at genus level. 

### 2.5. Statistical Analysis

The statistical processing of the data obtained from each evaluation was implemented by means with Statgraphics Plus version 5.1 (AV Trading, Umex, Dresden, Germany). For each replication, the mean was calculated, and the data were log transformed. The Student’s *t*-test was performed for statistical analysis. Differences were evaluated as significant if *p* < 0.05; *p* < 0.01; *p* < 0.001. 

## 3. Results and Discussion

The antimicrobial activity of essential oil has been known for many centuries [26]. The total viable count (TVC) values for the tested groups of chicken thigh are shown in Figure 1. The initial TVC value of chicken thigh was 3.43 ± 0.33 log cfu/g on day 0. A similar result was observed by Ismail et al. [27], who reported mean TVC populations of 3.32–5.77 log cfu/g for various raw and processed chicken products. We observed the highest TVC, 4.58 ± 0.48 log cfu/g, in APCS on day eight, and the lowest TVC, 3.09 ± 2.39 log cfu/g, was found in VPC samples after day 16 of storage at 4 ± 0.5 °C. In samples treated with essential oils, the lowest TVC, 3.43 ± 0.33 log cfu/g, was found on day 0. For VP + F samples, the highest TVC, 4.57 ± 0.30 log cfu/g, was found on day 16; for VP + S samples, the highest TVC, 4.42 ± 0.41 log cfu/g, was found on day 1. Statistically significant differences (*p* < 0.05) were found between APCS and VPC, APCS and VPRO, APCS and VP + F, VPC and VP + S, VPC and VP + F, and VPRO and VPC. Rapeseeds oil treatments did not affect the total viable count (TVC).

Our results were in agreement with the findings of Cosby et al. [28], Branen and Davidson [29], and Belfiore [30], who did not find an effect of a disodium ethylenediametetra-acetate (EDTA) and nisin (NIS) combination and storage under modified atmosphere packaging (MAP) or vacuum packaging (VP) on the TVC in their study. Dawson et al. [31] reported a reduction in growth of aerobic bacteria by 1–1.5 log cfu/g in ground chicken meat after 14 days of storage under modified atmosphere packaging. The data of the study Adiguzel et al. [32] clearly indicated that the essential oil of *Satureja hortensis* plant has strong antimicrobial activity against both bacteria and fungi, since 25 bacteria, eight fungi, and one yeast species were inhibited. In the study by Mahboubi and Kazempour [33] thymol, p-cymene, γ-terpinene, and carvacrol were the main components of *S. hortensis* oil, and thymol played an important role in providing antimicrobial activity. Many authors reported that fennel essential oil has good antimicrobial properties [14,34], while Miquel et al. [35] found that fennel essential oils exhibited very low antimicrobial activity. Our results of the TVC are in agreement with what has been previously reported and have shown that the treatment was successful for VP+F.

The processing and storage of meat prior to consumption can have a significant effect on meat quality [36]. Members of the *Enterobacteriaceae* on raw beef, lamb, pork, and poultry products, as well as on offal [37], were identified in the previous study [38]. In our study *Enterobacteriaceae* as a hygiene indicator [39], counts on day 0 were 3.75 ± 0.46 log cfu/g (Figure 2). On day 16 of storage, *Enterobacteriaceae* reached 4.87 ± 0.04 log cfu/g in APCS samples. In the case of VPC, the count of *Enterobacteriaceae* was lowest at 3.75 ± 0.46 log cfu/g on day 0 and highest at 4.88 ± 0.01 log cfu/g on day eight. The counts of *Enterobacteriaceae* in VPRO was highest at 0.43 ± 1.06 log cfu/g on day 16 and lowest at 3.75 ± 0.46 log cfu/g on day 0.

For VP + F, the lowest and highest *Enterobacteriaceae* counts were from 2.55 ± 1.99 log cfu/g on day 16 to 3.75 ± 0.64 log cfu/g on day 0. For VP + S, the lowest and highest counts were from 0.67 ± 1.03 log cfu/g on day eigh to 3.75 ± 0.64 log cfu/g on day 0. Statistically significant differences (*p* < 0.05) were found between all tested groups without VP + F, VP + S, APCS, and VPC.

The essential oils and their components are known because of their activity against a wide variety of microorganisms including gram-negative and gram-positive bacteria. There were found out that gram - negative bacteria are more resistant to the antagonistic effects of essential oils than gram-positive ones. It can be connected with the structure of the outer membrane, where no lipopolysaccharide are present. *F. vulgare* seed extracts and oil are rich in trans-anethole and other compounds, and they have positive effect against *C. albicans*, *E. coli*, *P. putida*, and other similar organisms [14]. Mihajilov-Krstev et al. [40] reported that *E. coli* O157:H7, *S. typhimurium*, *S. aureus* and *L. monocytogenes* are inhibited with *S. hortensis* essential oil. Essential oils of *S. hortensis* show wide spectrum and inhibition effect against growth of the human and phytopatogenic and food spoilage bacteria, fungi and yeasts species.

Lactic acid bacteria (LAB), a facultative anaerobe, are recognized as an important competitor of other spoilage related microbial groups under VP/modified atmosphere packaging (MAP) conditions [38,41]. The results of Ntzimani [42] indicate that LAB are an important part of precooked chicken microflora, irrespective of the packaging conditions and the antimicrobial treatment combination. This could contribute to their rapid growth between days 0 and two of storage, similar to LAB growth in beef stored under MAP at 5 °C [43]. Dominant LAB bacterial species in pre-cooked chicken product stored in air and MAP are reported as being the significant part of natural microflora of freeze-chilled chickens with number of 6.1–7.5 log cfu/g [44,45].

The initial number of LAB counts (Figure 3) was 3.95 ± 0.18 log cfu/g (day 0). The number of LAB in AC ranged from 3.45 ± 0.46 log cfu/g on day 16 to 4.06 ± 0.08 log cfu/g on day four. In the case VPC, the highest count of LAB 4.31 ± 0.29 log cfu/g was marked on day 16 of storage, and the lowest 3.69 ± 0.08 log cfu/g on day 12. In VPRO, VP+F, and VP+S, the highest LAB counts of 3.95 ± 0.18 log cfu/g were observed on day 0, and the lowest comprised 3.35 ± 0.21 on day 12, 3.39 ± 0.48 on day 16, and 3.28 ± 0.39 log cfu/g on day eight for each group, respectively. 

Statistically significant differences (*p* < 0.05) were found between AC and VPRO, AC and VP + S, VPC and VP + S, VPC and VP + F, and VPRO and VPC.

Different essential oils, such as oregano, were applied to control the growth of LAB. Oregano oil with a concentration of 1% was more effective than the concentration of 0.1% in regards to reducing the populations of LAB, while the combination of (modified atmosphere packaging) MAP and oregano oil at a concentration 1% had the greatest effect, as described by Chouliara et al. [2]. Kivanc et al. [46] reported that oregano and its essential oil inhibited the growth of *Lactobacillus plantarum*, but oregano spice stimulated *Lactobacillus plantarum* acid production. *Lactobacillus plantarum* in a liquid medium stimulated its growth and acid production of cumin at concentration 0.5, 1.0 and 2.0 % (*w*/*w*). *Lactobacillus plantarum* with acid production is inhibited with essential oil from cumin in high concentration (300 and 600 ppm). In our study, the count of *Pseudomonas* spp. was found only on day 0, with a number of 3.43 log cfu/g in all samples.

Together, 15 genera were identified, and they were *Aeromonas*, *Aromatoleum*, *Buttiauxella*, *Clostridium*, *Enterobacter*, *Hafnia*, *Lactobacillus*, *Lysinobacillus*, *Pantotea*, *Pseudomonas*, *Rahnella*, *Raoultella*, *Serratia*, *Staphylococcus*, and *Yersina*. In our study, only microorganisms identified with a score of more than two were mentioned. A total of 239 colonies were identified with this score for bacterial species identification. From fifteen genera, 46 species were isolated. Within the *Aeromonas genus*, *A. veronii* was the most frequently isolated, while *Clostridium novyi* (4%) was isolated less frequently (Table 1). Among LAB, only *Lactobacillus* spp. were isolated with 12 species. Genus *Pseudomonas* was represented with 9 species, and *P. flurorescens* (80%), *P. gessardii* (80%), and *P. proteolitica* (80%) were among the most abundant. *Enterobacteriaceae* were comprised of *Buttiauxella*, *Enterobacter*, *Hafnia*, *Rahnella*, *Raoultella*, and *Serratia*, with *Rahnella aquatis* (48%) and *Serratia plymuthica* (48%) being the most abundant (Table 1). The similar results of the chicken meat after treatment with essential oils were found in a study by Kačániová et al. [47].

## 4. Conclusions

The essential oils of *Feniculum vulgare* (fennel) and *Satureja hortensis* (savory) as natural food preservatives with potential antimicrobial activity in food industry were studied in our study. The use of fennel oil and savory oil was the most effective against the growth of the total viable count of lactobacilli and *Enterobactericeae* in this study. Based on microbiological analyses, treatment with fennel and savory oil prolongs the shelf-life of chicken thighs, in comparison to control samples.

## Figures and Tables

**Figure 1 microorganisms-07-00134-f001:**
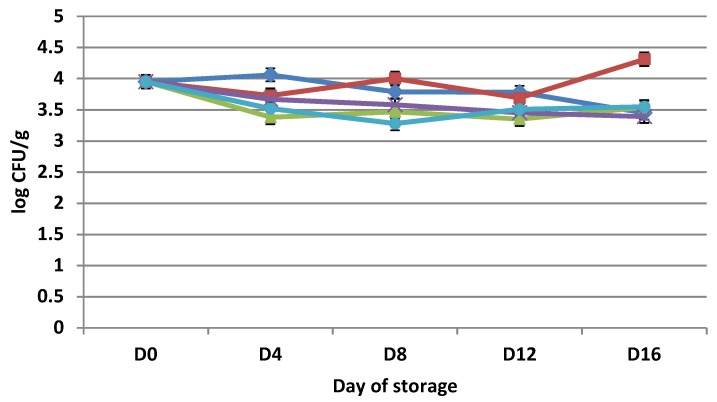
Changes (log cfu/g) in population of Total Viable Count (TVC) in chicken thigh stored in air (APCS, ♦); stored under vacuum (VPC, ■); stored under vacuum packaging with rapeseed oil (VPRO, ▲); stored under vacuum packaging with *Foeniculum vulgare* 0.2% essential oil (VP + F, **×**); and stored under vacuum packaging with *Satureja hortensis* 0.2% essential oil (VP + S, ●). Each point is the mean of three samples taken from two replicate experiments (*n* = 3 × 2 = 6). Error bars show SD.

**Figure 2 microorganisms-07-00134-f002:**
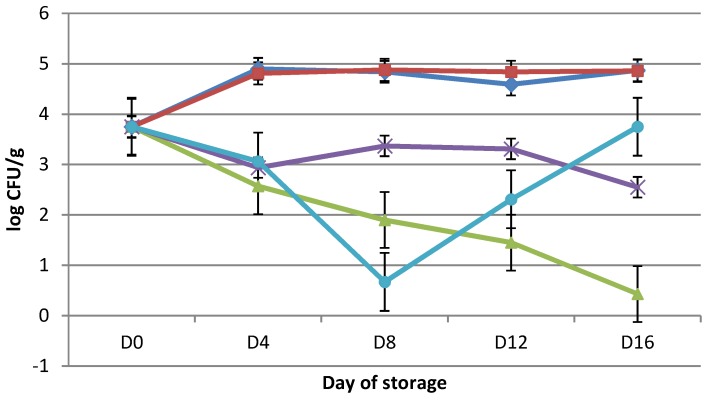
Changes (log cfu/g) in population of *Enterobacteriacae* in chicken thigh stored in air (APCS, ♦); stored under vacuum (VPC, ■); stored under vacuum packaging with rapeseed oil (VPRO, ▲); stored under vacuum packaging with *Foeniculum vulgare* 0.2% essential oil (VP + F, **×**); stored under vacuum packaging with *Satureja hortensis* 0.2% essential oil (VP + S, ●). Each point is the mean of three samples taken from two replicate experiments (*n* = 3 × 2 = 6). Error bars show SD.

**Figure 3 microorganisms-07-00134-f003:**
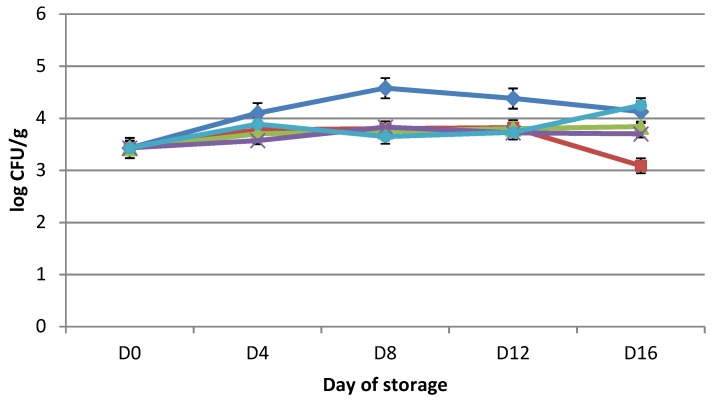
Changes (log cfu/g) in population of lactic acid bacteria (LAB) in chicken thigh stored in air (APCS, ♦); stored under vacuum (VPC, ■); stored under vacuum packaging with rapeseed oil (VPRO, ▲); stored under vacuum packaging with *Foeniculum vulgare* 0.2% essential oil (VP + F, **×**); stored under vacuum packaging with *Satureja hortensis* 0.2% essential oil (VP + S, ●). Each point is the mean of three samples taken from two replicate experiments (*n* = 3 × 2 = 6). Error bars show SD.

**Table 1 microorganisms-07-00134-t001:** Identification of bacteria isolated from chicken meat.

Species	No. of Bacterial Colonies/No. of Samples	Percentage %
*Aeromonas veronii*	16/25	64.00
*Aromatoleum alkani*	2/25	8.00
*Aromatoleum buckelii*	6/25	24.00
*Aromatoleum evansii*	9/25	36.00
*Buttiauxella gaviniae*	10/25	40.00
*Buttiauxella izardii*	4/25	16.00
*Buttiauxella warmboldiae*	6/25	24.00
*Clostridium difficile*	2/25	8.00
*Clostridium novyi*	1/25	4.00
*Enterobacter amnigenus*	2/25	8.00
*Enterobacter cloacae*	4/25	16.00
*Hafnia alvei*	3/25	12.00
*Lactobacillus amylophilus*	2/25	8.00
*Lactobacillus bifermentans*	3/25	12.00
*Lactobacillus casei*	5/25	20.00
*Lactobacillus fructivorans*	3/25	12.00
*Lactobacillus parabuchneri*	4/25	16.00
*Lactobacillus paracasei*	6/25	24.00
*Lactobacillus paraplantarum*	5/25	20.00
*Lactobacillus pentosus*	15/25	60.00
*Lactobacillus plantarum*	2/25	8.00
*Lactobacillus reuteri*	5/25	20.00
*Lactobacillus salivarius*	5/25	20.00
*Lactobacillus zeae*	6/25	24.00
*Lysinibacillus fusiformis*	4/25	16.00
*Pantoea ananatis*	2/25	8.00
*Pseudomonas anguilliseptica*	3/5	60.00
*Pseudomonas brenneri*	2/5	40.00
*Pseudomonas fluorescens*	4/5	80.00
*Pseudomonas fragi*	3/5	60.00
*Pseudomonas gessardii*	4/5	80.00
*Pseudomonas lundensis*	3/5	60.00
*Pseudomonas proteolytica*	4/5	80.00
*Pseudomonas resinovorans*	2/5	40.00
*Pseudomonas taetrolens*	6/25	24.00
*Rahnella aquatis*	12/25	48.00
*Raoultella ornithinolytica*	4/25	16.00
*Seratia fonticola*	5/25	20.00
*Serratia liquefaciens*	5/25	20.00
*Serratia odorifera*	9/25	36.00
*Serratia plymuthica*	12/25	48.00
*Staphylococcus capitis*	9/25	36.00
*Staphylococcus caprae*	5/25	20.00
*Staphylococcus epidermis*	6/25	24.00
*Staphylococcus pasteuri*	7/25	28.00
*Yersinia enterocolitica*	2/25	8.00

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
