# Peer review of "Combined Effect of Vacuum Packaging, Fennel and Savory Essential Oil Treatment on the Quality of Chicken Thighs"

_microorganisms, 2019, doi:10.3390/microorganisms7050134_

Round 1
Reviewer 1 Report
The article concerns the impact of vacuum packaging and the addition of essential oils from fennel and savory to chicken thighs. It is part of the same topics in the current search for alternatives to chemicals added to raw materials in order to prolong their durability. However, when we use oils, there are some problem with the lack of acceptance, among others fragrance, which is often referred to as too intense or unusual for the raw material/product.
Mainly comments on the content of the article:
1. I have reservations about the formulation of “Enterobacteraceae genera counts” (line 27). The term "Bacteria of the Enterobacteriaceae family" is more commonly used.
2. In line 27 the authors use the name Lactobacillus spp. Whereas in the methodology (line 113) they refer to “lactic acid bacteria”. After all, many bacteria with lactic acid production properties belong to the latter group, so only one type can not be isolated. The same applies is to discussing the results.
3. On line 28, the name "MALDI TOF MS Biotyper" is written differently than in line 116 ("MALDI-TOF MS Biotyper"). Please check this.
4. Lines 69-70: it is worth completing information about the used polyethylene bags with its properties: barrier/gas permeability.
5. Lines 73-75: there is no explanation as to why the rapeseed oil is introduced. As I guess, it's a medium for oils. Please, specify the purpose of its application, because in lines 147-148, the authors describe that it did not affect the number of TVC.
6. In line 79, there was no unit after digit 1 - the abbreviation "min" is missing.
7. Lines 80-83: Packing description - please re-edit, because the sentence in line 83 duplicates information from lines 80-81.
8. On what basis has the additive amount (0.2% v / w) of the oils been determined? Literature or previous experiences of authors? Please explain.
9. The whole structure of paragraph 2.1 should look a bit different, because it is difficult to imagine: first authors described variants of experience and finally the way of applying oils.
10. I am inclined to the opinion that the method of preparing the oils (chapter 2.2) should be presented before the preparation of samples (chapter 2.1), because first there were oils and only later they were used.
11. Line 90-91: please specify if the plant material for the production oils was fresh or dried.
12. Please carefully read the text “Results and discussion”, because there is a lack of space between the number and the unit (line 173). In the same line is the number "0-". What does the dash at the digit mean?
13. Line 202 and entire paragraphs and drawing related to LAB: see note No. 2.
14. Lines 216-219: at the end of the sentence there was no expression and it lost its meaning: "...... had the greatest effect Chouliara et al. [2]."
15. A certain surprise is the statement (not entirely legible) in lines 224-225: „In our study the count of Pseudomonas spp. was found only 0 day” In which samples? How big was the number?
16. In lines 226-237, the authors describe the results of the identification of microorganisms using MALDI-TOF MS Biotyper. I do not know this method and it is not clearly defined for me, what do the% of numbers mean? Is it about % compliance or any other meaning these values have? E.g. Clostridium novyi (4%) - how to interpret it? Perhaps it is worth entering a description of the interpretation of results in chapter 2.4.
17. Did the authors perform acceptance tests of the sensory smell of oil samples? It is worth taking into account in the next publication.
Author Response
RE: "Combined effect of vacuum packaging, fennel and savory essential oil
treatment on the quality of chicken thighs"
Manuscript ID: microorganisms-491546
We would like to thank the reviewers for their valuable comments and recommendations. The manuscript has been corrected in line with the comments of reviewers. All corrections are highlighted. English was improved throughout the manuscript.
1. I have reservations about the formulation of “Enterobacteriaceae genera counts” (line 27). The term "Bacteria of the Enterobacteriaceae family" is more commonly used.
Bacteria of the Enterobacteriaceae family was replaced.
2. In line 27 the authors use the name Lactobacillus spp. Whereas in the methodology (line 113) they refer to “lactic acid bacteria”. After all, many bacteria with lactic acid production properties belong to the latter group, so only one type cannot be isolated. The same applies is to discussing the results.
Lactic acid bacteria LAB was replaced.
3. On line 28, the name "MALDI TOF MS Biotyper" is written differently than in line 116 ("MALDI-TOF MS Biotyper"). Please check this.
It was corrected in all text.
4. Lines 69-70: it is worth completing information about the used polyethylene bags with its properties: barrier/gas permeability.
It was added: bags for food packaging with low permeability, with the gas and vapour protection.
5. Lines 73-75: there is no explanation as to why the rapeseed oil is introduced. As I guess, it's a medium for oils. Please, specify the purpose of its application, because in lines 147-148, the authors describe that it did not affect the number of TVC.
It was added: Rapeseed oil with any microorganisms was used as a solvent for concentration of essential oils. Rapeseed oil has good properties and it is safely for human health.
6. In line 79, there was no unit after digit 1 - the abbreviation "min" is missing.
It was added.
7. Lines 80-83: Packing description - please re-edit, because the sentence in line 83 duplicates information from lines 80-81.
It was described.
8. On what basis has the additive amount (0.2% v / w) of the oils been determined? Literature or previous experiences of authors? Please explain.
The additive amount 0.2% v/w of Eos been determined by our experience, because higher concentration very influent sensorial analyses of meat.
9. The whole structure of paragraph 2.1 should look a bit different, because it is difficult to imagine: first authors described variants of experience and finally the way of applying oils.
It was changed.
10. I am inclined to the opinion that the method of preparing the oils (chapter 2.2) should be presented before the preparation of samples (chapter 2.1), because first there were oils and only later they were used.
It was changed.
11. Line 90-91: please specify if the plant material for the production oils was fresh or dried.
It was added, dried medicinal plants.
12. Please carefully read the text “Results and discussion”, because there is a lack of space between the number and the unit (line 173). In the same line is the number "0-". What does the dash at the digit mean?
It was changed.
13. Line 202 and entire paragraphs and drawing related to LAB: see note No. 2.
It was changed.
14. Lines 216-219: at the end of the sentence there was no expression and it lost its meaning: "...... had the greatest effect Chouliara et al. [2]."
It was added as described Chouliara et al.
15. A certain surprise is the statement (not entirely legible) in lines 224-225: „In our study the count of Pseudomonas spp. was found only 0 day” In which samples? How big was the number?
It was added, number was 3.43 log cfu/g in all samples.
16. In lines 226-237, the authors describe the results of the identification of microorganisms using MALDI-TOF MS Biotyper. I do not know this method and it is not clearly defined for me, what do the% of numbers mean? Is it about % compliance or any other meaning these values have? E.g. Clostridium novyi (4%) - how to interpret it? Perhaps it is worth entering a description of the interpretation of results in chapter 2.4.
This results are obtained from calculation of previously column 1/25 where 2 is frequency from 25 samples ant it is 2:25=4%.
17. Did the authors perform acceptance tests of the sensory smell of oil samples? It is worth taking into account in the next publication.
The authors performed test of sensorial analyses. Panellists were asked to evaluate taste, smell, juiciness and tenderness of the samples. And this results are not publishing in this manuscript because journal aim is about microorganisms and test was not provide for full time because microbial characteristics show not very good results for sensorial analysis.

Reviewer 2 Report
The study is novel as it evaluates how fennel and savory essential oils can impact bacterial counts and growth in fresh chicken thighs under vacuum packaged conditions. This is an important area of research as consumers are becoming more interested in the safety and antimicrobial ingredients used in their foods. The manuscript is well written and provides data that support their conclusions and the need for additional research with fennel and savory essential oils.
Specific comments:
Abstract
Line 29 - change (0.05<P) to (P<0.05). This way it is written reads there was not statistical significance.
Introduction
Line 37 - Remove "of"
Line 48 - Add (fennel) after the scientific name.
Line 59 - Was the aim to examine the combined effects of the essential oils? The treatments and data only show fennel or savory essential oils not the combination of both.
Materials and methods
Line 67-68 - It should be added that each experiment was replicated 2 times. This is not ever written in the methods but is in the figure titles.
Lines 69 - 79 - Suggest changing "chicken thigh fresh meat" to "fresh chicken thigh meat"
Line 84-85 - what volume of solution was used?
Line 86 - is it Bouth or Both?
Line 88 - Is it chapter 2.2 or section 2.2?
Line 133-136 - What was the statistical design of the experiment and did the authors test for the affect of storage time or only treatment?
Conclusion
Line 241 - suggest adding fennel and savory after the scientific names.
Author Response
RE: "Combined effect of vacuum packaging, fennel and savory essential oil
treatment on the quality of chicken thighs"
Manuscript ID: microorganisms-491546
We would like to thank the reviewers for their valuable comments and recommendations. The manuscript has been corrected in line with the comments of reviewers. All corrections are highlighted. English was improved throughout the manuscript.
Line 29 - change (0.05<P) to (P<0.05). This way it is written reads there was not statistical significance.
It was changed.
Line 37 - Remove "of"
It was removed.
Line 48 - Add (fennel) after the scientific name.
It was added.
Line 59 - Was the aim to examine the combined effects of the essential oils? The treatments and data only show fennel or savory essential oils not the combination of both.
It was removed. It was mean as combination effect of oil and vacuum packaging.
Line 67-68 - It should be added that each experiment was replicated 2 times. This is not ever written in the methods but is in the figure titles.
It was added.
Lines 69 - 79 - Suggest changing "chicken thigh fresh meat" to "fresh chicken thigh meat"
It was changed.
Line 84-85 - what volume of solution was used?
It is in next sentence 0.2%.
Line 86 - is it Bouth or Both?
It is both and was changed.
Line 88 - Is it chapter 2.2 or section 2.2?
It was changed and it is chapter.
Line 133-136 - What was the statistical design of the experiment and did the authors test for the affect of storage time or only treatment?
The statistically design of the experiment were effect of treatment during storage.
Line 241 - suggest adding fennel and savory after the scientific names.
It was added.
